chemical engineering/physical chemistry/ mathematical modelling

carbon dioxide, corrosion mechanism, corrosion model

**Author for correspondence:**
Jigang Wang
e-mail: wangjigang9999@163.com

This article has been edited by the Royal Society of Chemistry, including the commissioning, peer review process and editorial aspects up to the point of acceptance.

# Mechanism and modelling of $CO_2$ corrosion on downhole tools

Jigang Wang[1], Lihui Meng[2], Zhenzhong Fan[2], Qingwang Liu[2] and Zhineng Tong[3]

[1]Guangdong University of Petrochemical Technology, Maoming, Guangdong, 525000, People's Republic of China
[2]Key Laboratory of Improving Oil Recovery by Ministry of Education, Northeast Petroleum University, Daqing, Heilongjiang 163111, People's Republic of China
[3]China Oilfield Services Ltd, No. 1581 Haichuan Road, Tanggu Marine High-tech Development Zone, Tianjin 300459, People's Republic of China

JW, 0000-0002-8845-0406

The conditions surrounding oil and gas exploration are becoming more hazardous, especially in oil and gas fields with a high quantity of corrosive components such as $CO_2$. $CO_2$ causes localized corrosion of tools made from metal, rubber and other materials in humid environments; this leads to corrosion failure in metal equipment and downhole tools such as drills pipes, casings and oil pipes, thereby reducing their service life. In this study, the composition, lattice and crystalline forms of corrosion products and corroded materials were analysed using scanning electron microscopy, energy spectrum analysis, X-ray diffraction and polarization curves, in order to investigate the corrosion mechanisms and influential factors for several common tool materials. A $CO_2$ corrosion model was established for two materials and the results were verified with optimal prediction values.

## 1. Introduction

Carbon dioxide flooding is one of the most effective technologies in modern oil extraction; it makes a positive contribution to the geological storage of carbon and has been rapidly developed [1,2]. However, $CO_2$ produces carbonic acid under suitable pressure and humidity conditions in the well. This is highly corrosive to cement and oil casings, causing severe corrosion to metal pipes and equipment, ultimately shortening their service life; this might result in vast economic loss and even leakage [3,4]. Research institutes in China and abroad have been investigating the corrosion mechanism of corrosive components, such as hydrogen sulfide [5,6] and carbon dioxide [7,8], and they have carried out a series of studies in recent decades. The conditions surrounding

the current development in China's oil and gas fields mean that it is critical to investigate the corrosion mechanism, corrosion predictions and corresponding protective measures of oil well pipes in harsh environments [9]. $CO_2$ corrosion is one of the most prevalent factors contributing to the damage of metal materials during oil and gas well exploitation and transportation worldwide. Therefore, numerous studies have explored the behaviour of carbon dioxide and the mechanism of carbon dioxide corrosion of low alloy steel pipes [10,11].

The corrosion of steel equipment by $CO_2$ can be divided into two categories: general corrosion (also known as uniform corrosion) and localized corrosion. Research has shown that $CO_2$ corrosion is caused by a series of complex physical and chemical processes that are influenced by temperature, $CO_2$ concentration [12] and pH [13]. $CO_2$ corrosion can be reduced by the formation of a thin film (nm) of organic inhibitors on the surface of the steel, or a protective corrosion product layer such as iron carbonate. These processes form a protective barrier on the metal surface based on the mechanism of adhesion to the metal surface [14].

In the 1980s, research into $CO_2$ corrosion focused on the effects of environmental and material factors on corrosion rates [15]. The main type of corrosion studied during this period was uniform corrosion; it was generally believed that the corrosion product film, formed on the surface of a material during the corrosion process [16], and flow rate were the decisive factors affecting the corrosion rate and corrosion morphology, while temperature was a major factor affecting the corrosion rate and the density of the corrosion product film [17]. Over time, the focus of $CO_2$ corrosion research shifted to the corrosion product film and multiphase flow. Studies of corrosion product films mainly concentrated on their effect on the mass transfer process, and their correlation with the corrosion rate and corrosion form (such as localized corrosion and pitting corrosion) [18,19]. Meanwhile, research into multiphase gas, water and oil was concerned with the effects of flow rate and flow pattern on the corrosion mass transfer process [20–22]. Furthermore, it was noted that in a $CO_2$-rich environment, galvanic corrosion caused by the coupling of carbon steel and other metals (or alloys) occurs alongside the corrosion damage caused by carbon steel itself [23–25]. Although stainless steel is resistant to $CO_2$ corrosion, it produces serious galvanic corrosion once coupled with alloy materials of higher potential in a $CO_2$ corrosive environment; this results in corrosion failure of the material [26]. Therefore, during the exploration of oil and gas fields containing $CO_2$, great attention should be paid to galvanic corrosion caused by the coupling of different metals [27,28].

In this study, the dynamic weight loss method was employed to evaluate $CO_2$ corrosion. The oil-field $CO_2$ corrosion evaluation standard was applied to the indoor experiment in order to measure the corrosion degree of N80 steel and a variety of anti-corrosion materials prepared using nano-, Ni-P plating and nitriding treatments. Based on the experimental results, the materials with good corrosion resistance were screened through performance characterization at various temperatures, pressures and flow rates by simulating the actual corrosion environment. The predictions and verification of the corrosion rate model were carried out by modifying the semi-empirical formula from the de Waard model [29–31] based on the experimental and field test conditions.

# 2. Material and methods

## 2.1. Equipment and samples

The experiments were conducted using the following equipment: a high temperature and pressure dynamic reaction kettle with a temperature control range of 0–200°C and control accuracy of $\pm 1$°C; a vacuum drying oven with a temperature control range of 0–200°C; $CO_2$ gas cylinders; an analytical balance with accuracy of 0.0001 g; a Quantum 600F scanning electron microscope (Oxford Instruments, UK); an Oxford INCA x-act energy spectrum analyser (Oxford Instruments, UK); an X-ray diffractometer; a PCI4/750 electrochemical comprehensive test system; absorbent cotton, gauze and bamboo tweezers; and solutions and film-removing solution (the main components of film-removing solution: hexamethylenetetramine, potassiumthiocyanate, nitric acid, aniline and distilled water). Test materials included polytetrafluoroethylene; nano-, nitride and nickel-plate-treated materials; and N80 and J55 steels. A schematic diagram of the experimental equipment and the original forms of the samples are shown in figure 1.

## 2.2. Experimental procedures

First, the film-removing solution was prepared in house. Next, the samples were treated in a series of steps that included soaking in 120# solvent oil, drying and measuring the surface area and weight.

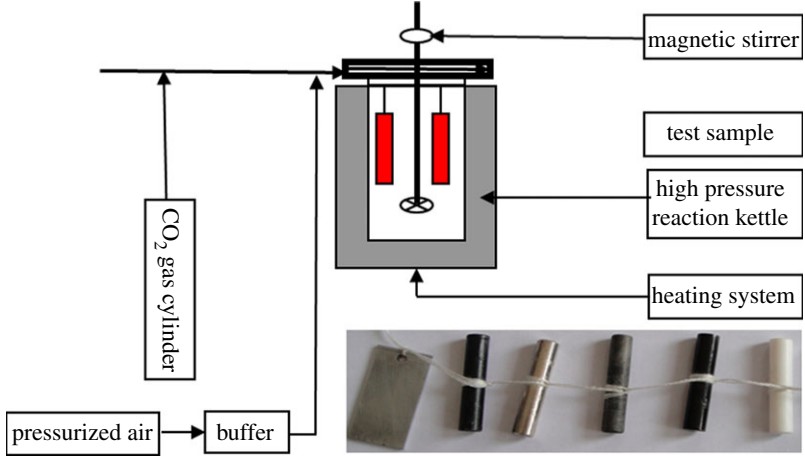

**Figure 1.** High temperature and pressure dynamic reaction kettle and the original forms of the test samples.

The samples were then suspended from a nylon string in a high-pressure container which was filled with water and bubbled with $CO_2$ (more than 2 h). The container was then closed, pressurized and heated. The process was observed and recorded. After 72 h, the samples were taken out, wiped with the film-removing solution, and soaked in an ethanol solution for 1 h. Finally, the test samples were dried at 110°C for 2 h in a constant-temperature drying oven before they were cooled in a desiccator and weighed.

## 2.3. Effect of partial pressure on corrosion rate at different temperatures

Based on the actual working conditions in oil and gas wells, where the well head temperature is 20–30°C and the well head partial pressure of $CO_2$ is 0.1–5.0 MPa, the corresponding temperature and pressure were calculated for different depths of the well shaft according to a temperature and pressure gradient algorithm; these were used as the experimental parameters. A high-pressure container was used for the corrosion experiment with different materials at various temperatures and $CO_2$ partial pressures. The test duration was 72 h.

## 2.4. Performance analysis

Scanning electron microscopy (SEM) was used to observe changes in the surface micromorphology of the samples before and after corrosion. An energy dispersive spectroscopy (EDS) analysis was carried out on various regions of the sample surface before and after an acid wash, and the corrosion products were analysed using X-ray diffraction to measure the weight of individual elements. The polarization curve was measured using the PCI4/750 electrochemical comprehensive test system. The electric potential scanning range was $\pm 250$ mV, and the scanning speed was $1$ mV s$^{-1}$. The electrochemical test area was $1$ cm$^2$, and the composition of the solution was prepared to simulate a formation water solution.

# 3. Results and discussion

## 3.1. Effect of partial pressure on corrosion rate at various temperatures

The effect of partial pressure was examined at 20, 50, 80 and 110°C, as shown in figure 2. The results show that polytetrafluoroethylene hardly corrodes under any temperature and pressure conditions, while the corrosion rates of all the other materials increase in response to increasing temperature and pressure. The corrosion rates of all the test materials are in the order of: $R_{N80} > R_{J55} > R_{nitride} > R_{nano} > R_{nickel-plated}$. When the temperature was below 50°C and the pressure was less than 5.0 MPa, there was minimal corrosion of the test pieces, as shown in figure 2a,b. However, when the pressure was greater than 5.0 MPa, major corrosion took place; among the metal materials, this occurred at the highest rate on N80 and lowest the rate on the nickel-plated material. Figure 2c,d illustrates that relatively severe corrosion of the nano- and nitride-treated materials, as well as the N80 and J55 steels occurred when the testing temperature was above 80°C and the pressure was greater than 3.0 MPa. Of the metal materials, N80 had the highest corrosion rate, while nickel-plated material still had a relatively low corrosion rate. Therefore,

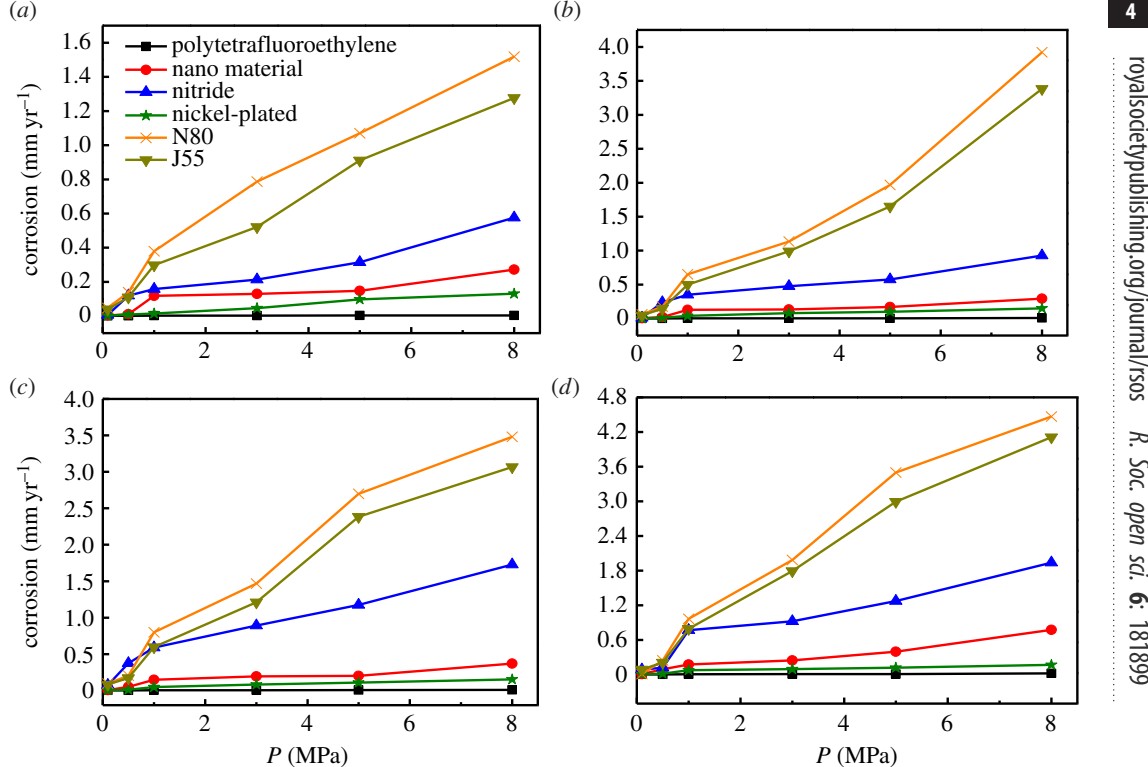

**Figure 2.** Effect of partial pressure on corrosion rate at (*a*) 20°C, (*b*) 50°C, (*c*) 80°C and (*d*) 110°C.

**Table 1.** Composition of the N80 and J55 steels used in this study.

| materials | C (%) | Si (%) | Mn (%) | P (%) | S (%) | Cr (%) | Ni (%) | Cu (%) |
|---|---|---|---|---|---|---|---|---|
| J55 | 0.34−0.39 | 0.2−0.35 | 1.25−1.5 | ≤0.02 | ≤0.015 | ≤0.15 | ≤0.2 | ≤0.2 |
| N80 | 0.34−0.38 | 0.2−0.35 | 1.45−1.70 | ≤0.02 | ≤0.015 | ≤0.15 | | |

we subsequently focused our analysis on the micromorphology of four test materials: nickel-plated, nitride treated, N80 and J55 steel post $CO_2$ corrosion.

## 3.2. SEM and energy spectrum analysis of corrosion products

The chemical compositions of the N80 and J55 steels used in this study are listed in table 1. The electron micrographs of the corrosion products from the four materials are shown in figure 3. The surface of the corrosion product film on J55 was relatively uniform, indicating that the corrosion product is in the form of particles, and shows signs of fluid erosion. These particles are approximately spherical and differ in size, with obvious pores among the crystalline spherical particles, which clearly shows the crystal growth steps. The surface of the nitride sample was covered with an even layer of corrosion product; the corrosion product was compact on the inner layer with a large quantity of loosely packed amorphous granular product on the outer layer. Although there were certain regular forms, there were no obvious crystal growth steps. Cracking dominated on the surface of the corrosion product film, while erosion and spalling occurred in certain localized areas. There was only tiny pitting corrosion on the metal base. Therefore, it can be concluded that the thick compact base layer of the corrosion product film substantially diminished the permeation of the corrosive liquid and therefore protected the metal base. The corrosion product of N80 was a form of platform erosion. The corrosion product film was very thick, but had a relatively weak bond with the base. Peeling between the base and the film occurred at relatively weak points, and thus, the corrosion product film had a distinct mesh structure, with the crystal growth of the corrosion product proceeding in a certain direction. There were a large number of white flocculent compounds around the crystalline corrosion product,

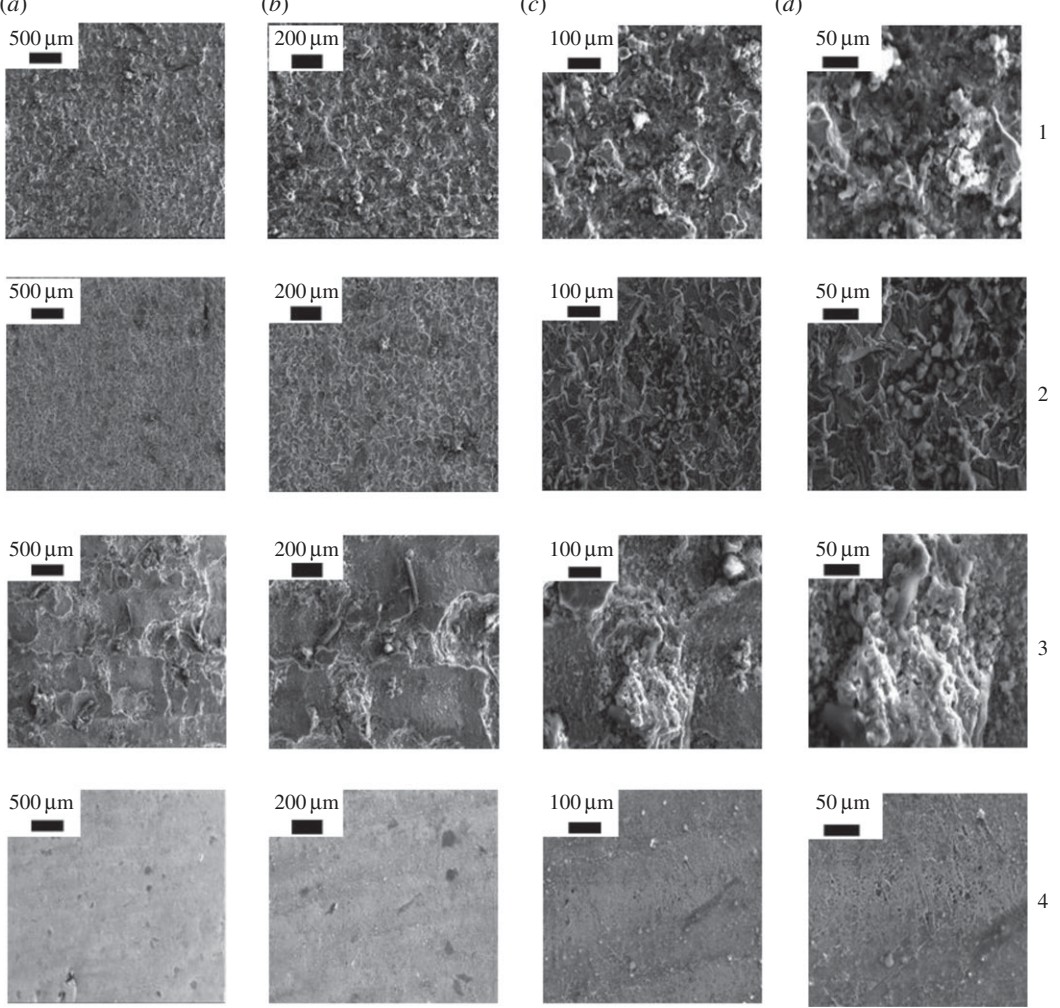

**Figure 3.** Surface SEM images of the (1) J55, (2) nitride, (3) N80 and (4) nickel-plated samples after corrosion (widths of (a) 500 μm, (b) 200 μm, (c) 100 μm and (d) 50 μm).

and numerous cracks and holes on the corrosion surface. The surface of the nickel-plated material was even, with minor corrosion of the metal surface. Thus, it was categorized as uniform corrosion and there was no large-scale corrosion. There was a small amount of pitting corrosion resulting from damage to the nickel-plating layer, which allowed the medium to contact the test material. Uniform corrosion occurred locally on the nickel-plated layer; however, the degree of corrosion was relatively low, and the corrosion product was washed away due to poor adhesion to the surface.

## 3.3. Energy spectrum analysis of corroded materials

From figure 4 and table 2, it can be seen that after corrosion, the J55 material showed a significant increase in the weight percentage and quantity of atoms of carbon and oxygen, based on spectra 1 and 2. Meanwhile, the weight percentage and atom quantity of iron were greatly reduced, indicating that carbon dioxide caused a strong corrosion of the material. Spectrum 3 shows that the weight percentage and atom quantity of carbon and oxygen are relatively small, and no weight percentage or atom quantity is given for the oxygen atoms. In addition, the weight percentage and atom quantity of iron are very high, revealing that the corrosion is minor. Energy spectrum analysis of spectrum 2 illustrates that the percentage of O, C, Fe and Mn atoms are up to 46.55%, 15.18%, 37.78% and 0.49%, respectively. Hence, we speculate that the crystal particles of this corrosion product are an Fe carbonate mainly composed of $FeCO_3$ and a double salt of $(Fe \cdot Mn)CO_3$ composed of $Fe^{2+}$ and $Mn^{2+}$.

Figure 4 and table 3 show the results of SEM and energy spectrum analysis for three representative points of the nitride materials. The weight percentages and atom quantities of carbon and oxygen at the three spectral points vary to a small extent. The corrosion rate is much lower than that of the J55 material. At the same time,

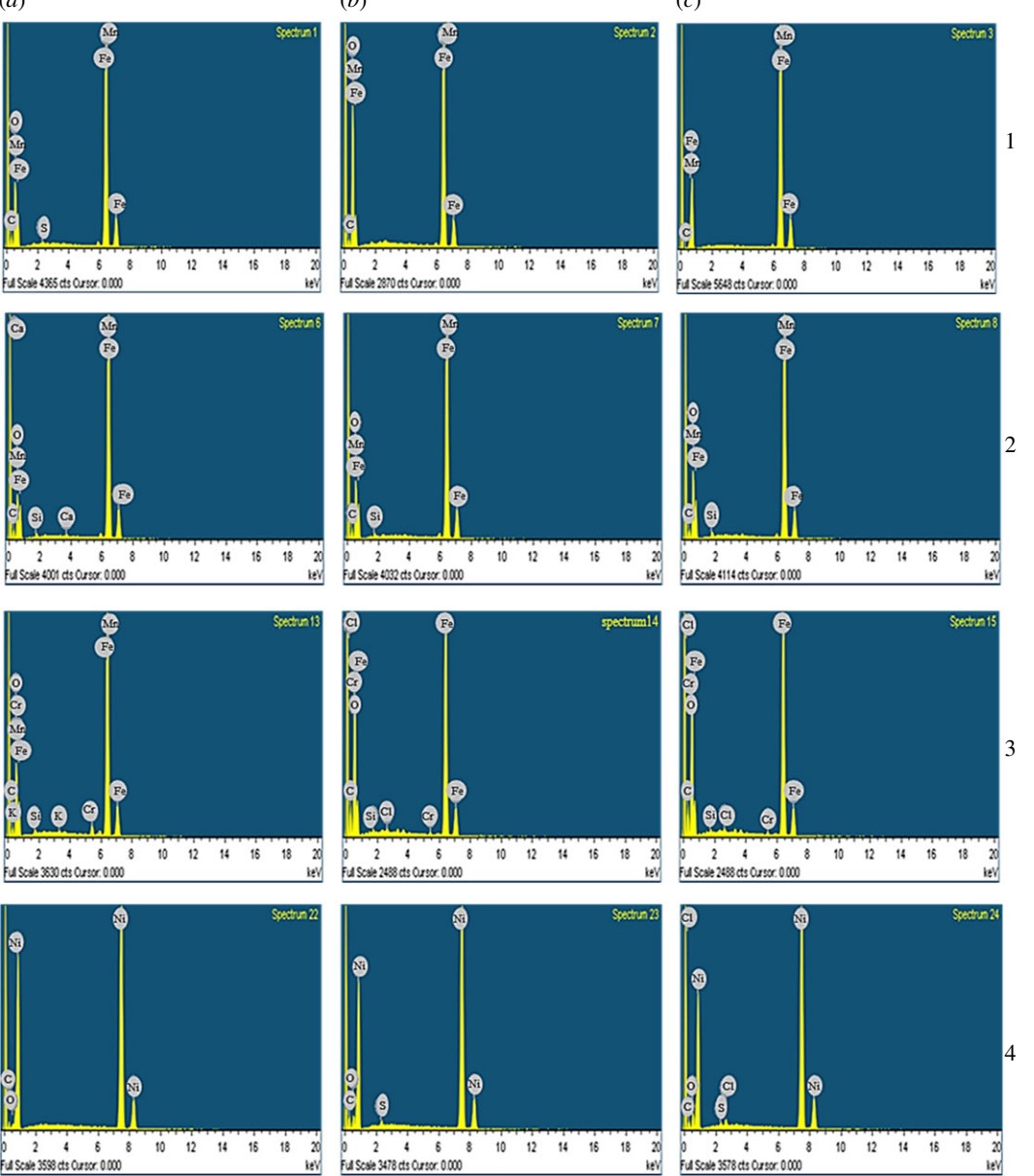

**Figure 4.** Energy spectra of (1) J55, (2) nitride, (3) N80 and (4) nickel-plated test materials after corrosion. ($a-c$) Are the three products with different corrosion degrees, respectively.

**Table 2.** Weight and atomic percentages of elements in corroded J55 material.

| element | spectrum *a* | | spectrum *b* | | spectrum *c* | |
|---|---|---|---|---|---|---|
| | weight (%) | atomic (%) | weight (%) | atomic (%) | weight (%) | atomic (%) |
| C | 9.39 | 25.81 | 5.95 | 15.18 | 2.44 | 10.40 |
| O | 13.91 | 28.71 | 24.31 | 46.55 | / | / |
| S | 0.28 | 0.28 | / | / | / | / |
| Mn | 1.24 | 0.74 | 0.87 | 0.49 | 1.25 | 1.17 |
| Fe | 75.19 | 44.45 | 68.86 | 37.78 | 96.31 | 88.43 |
| status | strong corrosion | | severe corrosion | | minor corrosion | |

**Table 3.** Weight and atomic percentages of elements in corroded nitride material.

| element | spectrum a | | spectrum b | | spectrum c | |
|---|---|---|---|---|---|---|
| | weight (%) | atomic (%) | weight (%) | atomic (%) | weight (%) | atomic (%) |
| C | 9.22 | 27.03 | 8.58 | 24.62 | 7.96 | 22.43 |
| O | 9.82 | 21.59 | 12.20 | 26.28 | 14.21 | 30.08 |
| Si | 0.43 | 0.54 | 0.31 | 0.38 | 0.49 | 0.59 |
| Ca | 0.29 | 0.26 | / | / | / | / |
| Mn | 1.26 | 0.81 | 0.98 | 0.61 | 1.13 | 0.70 |
| Fe | 78.98 | 49.78 | 77.93 | 48.10 | 76.21 | 46.21 |
| status | severe corrosion | | minor corrosion | | minor corrosion | |

**Table 4.** Weight and atomic percentages of elements in corroded N80 material.

| element | spectrum a | | spectrum b | | spectrum c | |
|---|---|---|---|---|---|---|
| | weight (%) | atomic (%) | weight (%) | atomic (%) | weight (%) | atomic (%) |
| C | 7.55 | 21.11 | 14.53 | 30.28 | 5.65 | 17.55 |
| O | 15.31 | 32.14 | 27.98 | 43.77 | 11.52 | 26.85 |
| Si | 0.35 | 0.42 | 0.26 | 0.23 | 0.34 | 0.45 |
| Cr | 1.73 | 1.12 | 0.35 | 0.17 | 1.33 | 0.96 |
| Mn | 0.82 | 0.50 | / | / | 0.74 | 0.50 |
| Fe | 74.01 | 44.51 | 56.58 | 25.35 | 80.42 | 53.70 |
| status | strong corrosion | | severe corrosion | | minor corrosion | |

**Table 5.** Weight and atomic percentages of elements in corroded nickel-plated material.

| element | spectrum a | | spectrum b | | spectrum c | |
|---|---|---|---|---|---|---|
| | weight (%) | atomic (%) | weight (%) | atomic (%) | weight (%) | atomic (%) |
| C | 6.96 | 26.35 | 11.28 | 36.39 | 7.64 | 26.66 |
| O | 0.78 | 2.21 | 2.74 | 6.64 | 3.67 | 9.61 |
| S | / | / | 0.37 | 0.45 | 0.24 | 0.32 |
| Cl | / | / | / | / | 0.53 | 0.62 |
| Ni | 92.26 | 71.44 | 85.61 | 56.52 | 87.93 | 62.79 |
| status | tiny corrosion | | minor corrosion | | minor corrosion | |

the weight percentage and atom quantity of iron only vary within a small range. It can be concluded that carbon dioxide causes uniform corrosion on the material, and that the corrosion rate is low.

Figure 4 and table 4 indicate that the corrosion product film of N80 contained several elements, including Fe, Cr, Mn, C and O, with the atomic percentages of O, C, Fe and Cr equal to 43.77%, 30.28%, 25.35% and 0.17%, respectively. We speculate that the crystal particles in this corrosion product may be composed of Fe carbonate, which mainly contains $FeCO_3$ and a double salt of $(Fe \cdot Cr)CO_3$ composed of $Fe^{2+}$ and $Cr^{3+}$. Comparison of the three figures shows that this test piece had pitting corrosion under these conditions, which was also found on the J55 sample.

Figure 4 and table 5 show that the weight percentage of carbon and oxygen, and the atomic quantity data of the three spectral points, varies only slightly after corrosion, especially in the case of oxygen. Similarly, the weight percentage and atomic quantity of nickel do not change substantially, and no

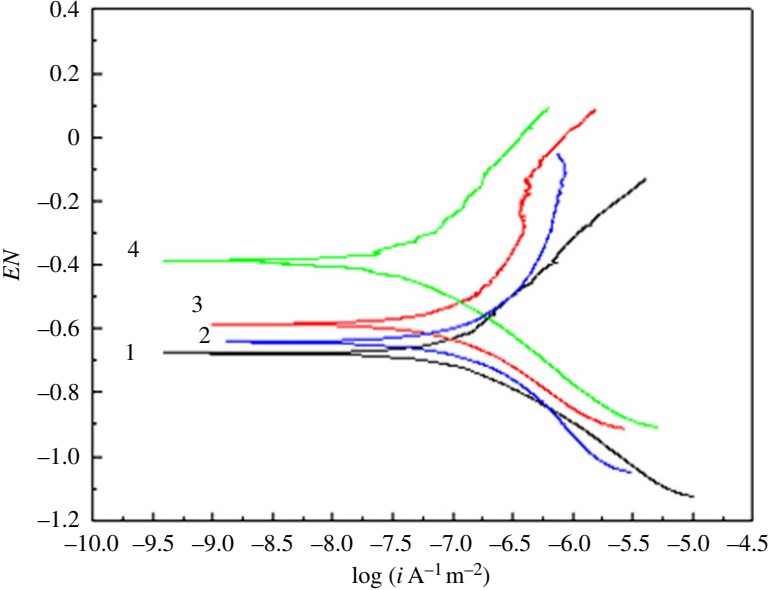

**Figure 5.** Diagram of polarization curves for (1) N80, (2) J55, (3) nitride and (4) nickel-plated samples.

iron is detected, which indicates a thick coating. The weight percentage of nickel is close to the range of its atom quantity, showing that carbon dioxide causes uniform corrosion of the material, and that the corrosion rate is very slow.

## 3.4. Analysis of polarization curves

Polarization curves, obtained using the PCI4/750 electrochemical comprehensive test system, are shown in figure 5.

Figure 5 illustrates that the corrosion potential and corrosion current of the N80, J55 and nitride samples are relatively close, while the corrosion potential of the nickel-plated sample is clearly positively shifted, and the corrosion current density is greatly reduced. This indicates that the corrosion resistance of the nickel-plated sample in the medium is better than that of the other three samples. Therefore, we established a corrosion model which predicted N80 to have the most severe corrosion and the nickel-plated material to have the best sustained release feature.

# 4. Establishment of the corrosion prediction model

There are many factors which affect the $CO_2$ corrosion characteristics of steel. In particular, temperature and $CO_2$ partial pressure ($PCO_2$) have a significant impact on $CO_2$ corrosion. The effect on corrosion rate is largely reflected by the impact of temperature on the formation of a protective film. Based on the indoor experimental data, a corrosion rate model was established and applied to a high temperature and pressure environment with a temperature range of 20–130°C and a $CO_2$ partial pressure of 12–28 MPa. Developed from the de Waard model, our model was established with modified constants according to the effects of temperature and pressure (table 6)

$$\lg V_c = a - \frac{2320}{t+273} - 5.55 \times 10^{-3} t + 0.67 \times \lg PCO_2,$$

where $V_c$ is the corrosion rate, mm yr$^{-1}$; $PCO_2$ is the partial pressure of $CO_2$, MPa; and $t$ is temperature, °C.

The constant $a$ is obtained through regression with temperature and pressure experimental data. The values of $a$ for N80 and the nickel-plated material are shown in figures 6–9, respectively. The value of $a$ for N80 shows a linear trend, so the fitted value was calculated by

$$a_{N80} = \frac{a_1 + a_2}{2}.$$

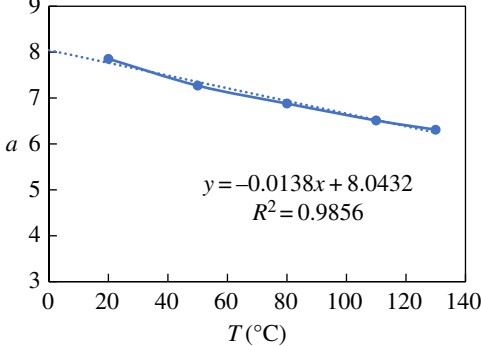

**Figure 6.** Fitting of constant $a_1$ and temperature.

$$y = -0.0138x + 8.0432$$
$$R^2 = 0.9856$$

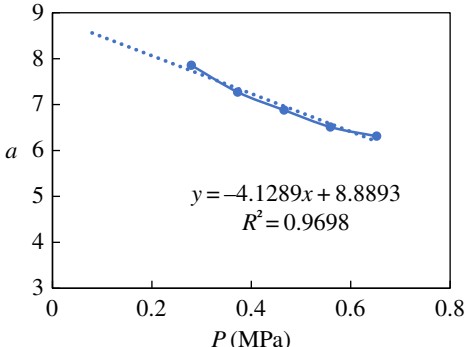

**Figure 7.** Fitting of constant $a_2$ and pressure.

$$y = -4.1289x + 8.8893$$
$$R^2 = 0.9698$$

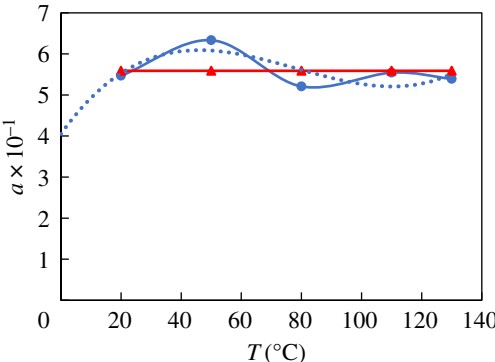

**Figure 8.** Fitting of constant $a_1$ and temperature.

**Table 6.** Experimental data of the corrosion rates of N80 and nickel-plated materials.

| temperature (°C) | $PCO_2$ (MPa) | corrosion rate of N80 material (mm yr$^{-1}$) | corrosion rate of nickel-plated material (mm yr$^{-1}$) |
|---|---|---|---|
| 20 | 0.2796 | 0.26573 | 0.00117 |
| 50 | 0.3728 | 0.25537 | 0.02882 |
| 80 | 0.3660 | 0.29177 | 0.03314 |
| 110 | 0.5592 | 0.25822 | 0.04149 |
| 130 | 0.6524 | 0.24788 | 0.05243 |

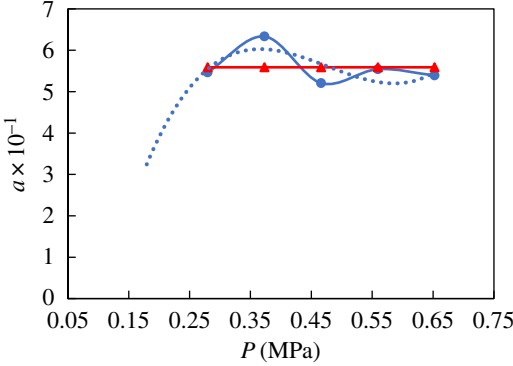

**Figure 9.** Fitting of constant $a_2$ and pressure.

**Table 7.** Comparison of analytical results from the $CO_2$ corrosion model.

| steel grade and pipe materials | group no. | temperature (°C) | P (MPa) | $PCO_2$ (MPa) | corrosion rate (mm yr$^{-1}$) predicted value | actual value |
|---|---|---|---|---|---|---|
| N80 | 1 | 40 | 17.5 | 0.408 | 0.28530 | 0.28899 |
| | 2 | 50 | 21.2 | 0.494 | 0.27454 | 0.28302 |
| | 3 | 60 | 24.1 | 0.562 | 0.26799 | 0.27485 |
| | 4 | 70 | 27.5 | 0.641 | 0.27691 | 0.26985 |
| | 5 | 80 | 30 | 0.699 | 0.29634 | 0.29892 |
| nickel-plated | 1 | 40 | 17.5 | 0.408 | 0.00641 | 0.00662 |
| | 2 | 50 | 21.2 | 0.494 | 0.01089 | 0.01206 |
| | 3 | 60 | 24.1 | 0.562 | 0.01718 | 0.01835 |
| | 4 | 70 | 27.5 | 0.641 | 0.02640 | 0.02750 |
| | 5 | 80 | 30 | 0.699 | 0.03833 | 0.03924 |

The value of $a$ for the nickel-plated material fluctuates around the average, so this value, $a = 0.589$, was applied to the model. The models were then established as below

$$\lg V_{\text{N80}} = \frac{a_1 + a_2}{2} - \frac{2320}{t + 273} - 5.55 \times 10^{-3}t + 0.67 \times \lg PCO_2$$

and
$$\lg V_{\text{nickel−plated}} = 0.589 - \frac{2320}{t + 273} - 5.55 \times 10^{-3}t + 0.67 \times \lg PCO_2.$$

According to the aforementioned analysis, the corrosion extrema appear around 60 and 110°C. Localized corrosion is more prominent between 60 and 110°C. When the temperature is lower than 60°C, corrosion tends to be uniform; when the temperature is at or above 110°C, $Fe_3O_4$ becomes the dominant proportion of the film. Since the $CO_2$ corrosion process occurs along with the depolarization of hydrogen, and it is completed by the hydronium ion in the solution, and because the hydrogen ion is formed by the breakdown of carbonic acid, corrosion is accelerated under high $CO_2$ partial pressure. This leads to the increased concentration of dissolved carbonic acid and the subsequent increase in the concentration of hydrogen ions released from the carbonic acid.

# 5. Example calculations from the corrosion prediction model

The data from five wells, provided by Block Fang 48, show that at downhole depths of 850 and 1730 m, the bottom hole pressures are approximately 21.2 and 30 MPa, respectively, while the bottom

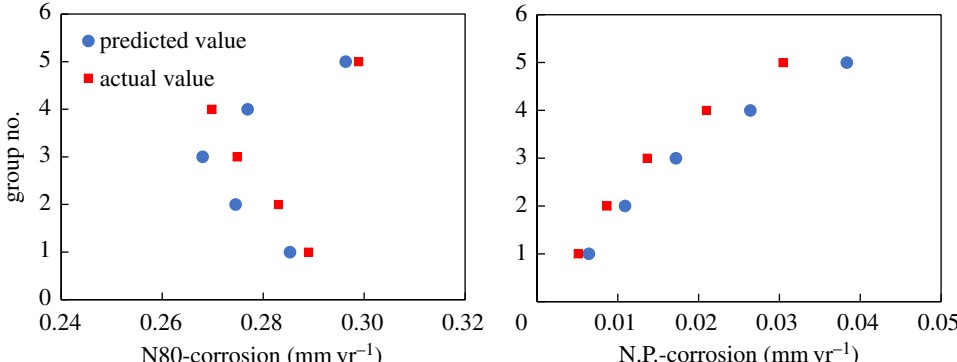

**Figure 10.** Comparison of corrosion rates of the N80 and the nickel-plated material.

temperatures are approximately 55 and 80°C, respectively. The actual corrosion rate of N80 and the nickel-plated material at a depth of 1730 m were analysed and compared with the simulated data. The detailed results are shown in table 7.

Table 7 and figure 10 indicate that the values predicted by the model are generally larger than the actual values detected. The value predicted for the N80 material is very close to the actual value with an error of approximately 2%, which indicates a decent prediction by the model. The value predicted for the nickel-plated material deviates from the actual value by approximately 6%, which shows a relatively good result.

## 6. Conclusion

We determined that the corrosion rates of various materials are in the order: $R_{N80} > R_{J55} > R_{nitride} > R_{nano} > R_{nickel\text{-}plated}$. Galvanic corrosion can be avoided to a large extent through comparison of the corrosion rates. In addition, we obtained the morphology and composition of the corrosion products of various materials using SEM, energy spectrum analysis, and X-ray diffraction. The corrosion products of J55 and N80 were relatively thick with the morphology forming over multiple steps, indicating severe corrosion. The base of the nitride material was protected by the dense corrosion product film that formed on its surface, and thus only experienced minor corrosion. The surface of the nickel-plated material was slightly corroded due to damage of the nickel-plating layer; this was categorized as minor corrosion. Meanwhile, the polarization curve clearly illustrated that the corrosion potential of the nickel-plated sample was positively shifted, and the corrosion current density was greatly reduced. The corrosion resistance of the nickel-plated material in the medium greatly improved the resistance of the downhole tools to $CO_2$ corrosion. By modifying the de Waard model, we established and verified a suitable corrosion rate model. The prediction results suggest a corrosion rate slightly higher than the actual corrosion rate for nickel-plated material. For the highly corrosive N80 material, the predicted value is relatively small. Compared with the nickel-plated material, the single factor of the experimental condition is more prominent than the actual condition, so the predicted value is relatively small. The prediction results have deviations of approximately 2 and 6% for the N80 and nickel-plated material, respectively. The model was able to make fairly good predictions, and can therefore help operators to take timely protective measures to reduce economic loss.

Data accessibility. Weight and atomic percentages of elements in corroded material data and corrosion rates data: data available from the Dryad Digital Repository: https://doi.org/10.5061/dryad.fn1r084 [32].

Authors' contributions. J.W.: drafting the article; substantial contributions to conception and design; agreement to be accountable for all aspects of the work in ensuring that questions related to the accuracy; L.M.: substantial contributions to conception and design; acquisition of data; analysis and interpretation of data; Z.F.: revising it critically for important intellectual content; Q.L.: final approval of the version to be published; Z.T.: integrity of any part of the work are appropriately investigated and resolved.

Competing interests. We declare we have no competing interests.

Funding. Each author was supported by the National Natural Science Foundation. Fund no. 51774089.

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
