## [Reviewer comments · Royal Society Open Science]

Review History

RSOS-181899.R0 (Original submission)

Review form: Reviewer 1

Is the manuscript scientifically sound in its present form?

Yes

Are the interpretations and conclusions justified by the results?

Yes

Is the language acceptable?

Yes

Is it clear how to access all supporting data?

Not Applicable

Do you have any ethical concerns with this paper?

No

Have you any concerns about statistical analyses in this paper?

No

Recommendation?

Accept with minor revision (please list in comments)

Comments to the Author(s)

See the attached file (Appendix A).

Review form: Reviewer 2

Is the manuscript scientifically sound in its present form?

Yes

Are the interpretations and conclusions justified by the results?

Yes

Is the language acceptable?

No

Is it clear how to access all supporting data?

Not Applicable

Do you have any ethical concerns with this paper?

No

Have you any concerns about statistical analyses in this paper?

No

Recommendation?

Major revision is needed (please make suggestions in comments)

Comments to the Author(s)

Overall paper presents mayrid experimental results with detailed and strong analysis. Please clarify a few minor points as listed below

1. In "experimental procedures", what is the film removing solution? What is it for?
2. Define corrosion rate. In line 21-23, 5 MPa was picked as the threshold pressure distinguishing between low and high corrosion rate. But comparing the corrosion rate at 5 MPa and 8 Mpa, there is not much difference according to figure 2
3. For Figure 4 energy spectra, provide some detailed on the spectra, especially the difference between a), b), and c). What are they referring to?
4. For table2-6, the definitions for severe vs minor vs strong corrosion are not clear
5. In the corrosion model, please explain a little more how the corrosion constants were deducted. For example, for nickel-plated, what is the Y-axis and units for figure 8-9
6. In line 22, it states "Table 7 and Figure 10 indicate that the values predicted by the model are generally larger than the actual values detected.". According to figure 10, the prediction values for N80 are smaller than actual values

Decision letter (RSOS-181899.R0)

06-Feb-2019

Dear Professor Wang:

Title: Mechanism and Modelling of CO₂ Corrosion on Downhole Tools
Manuscript ID: RSOS-181899

The editor assigned to your manuscript has now received comments from reviewers. We would like you to revise your paper in accordance with the referee and Subject Editor suggestions which can be found below (not including confidential reports to the Editor). Please note this decision does not guarantee eventual acceptance.

Please submit your revised paper before 01-Mar-2019. Please note that the revision deadline will expire at 00.00am on this date. If we do not hear from you within this time then it will be assumed that the paper has been withdrawn. In exceptional circumstances, extensions may be possible if agreed with the Editorial Office in advance. We do not allow multiple rounds of revision so we urge you to make every effort to fully address all of the comments at this stage. If deemed necessary by the Editors, your manuscript will be sent back to one or more of the original reviewers for assessment. If the original reviewers are not available we may invite new reviewers.

Please also include the following statements alongside the other end statements. As we cannot publish your manuscript without these end statements included, if you feel that a given heading is not relevant to your paper, please nevertheless include the heading and explicitly state that it is not relevant to your work.

- Ethics statement

Please clarify whether you received ethical approval from a local ethics committee to carry out your study. If so please include details of this, including the name of the committee that gave consent in a Research Ethics section after your main text. Please also clarify whether you received informed consent for the participants to participate in the study and state this in your Research Ethics section.

OR

Please clarify whether you obtained the necessary licences and approvals from your institutional animal ethics committee before conducting your research. Please provide details of these licences and approvals in an Animal Ethics section after your main text.

OR

Please clarify whether you obtained the appropriate permissions and licences to conduct the fieldwork detailed in your study. Please provide details of these in your methods section. Once again, thank you for submitting your manuscript to Royal Society Open Science and I look forward to receiving your revision. If you have any questions at all, please do not hesitate to get in touch.

On behalf of the Subject Editor Professor Anthony Stace and the Associate Editor Professor Hazel Cox.

RSC Associate Editor:
Comments to the Author:
(There are no comments.)

RSC Subject Editor:
Comments to the Author:
(There are no comments.)

Reviewers' Comments to Author:
Reviewer: 1

Comments to the Author(s)
See the attached File

Reviewer: 2

Comments to the Author(s)

Overall paper presents mayrid experimental results with detailed and strong analysis. Please clarify a few minor points as listed below

1. In "experimental procedures", what is the film removing solution? What is it for?
2. Define corrosion rate. In line 21-23, 5 MPa was picked as the threshold pressure distinguishing between low and high corrosion rate. But comparing the corrosion rate at 5 MPa and 8 Mpa, there is not much difference according to figure 2
3. For Figure 4 energy spectra, provide some detailed on the spectra, especially the difference between a), b), and c). What are they referring to?
4. For table2-6, the definitions for severe vs minor vs strong corrosion are not clear
5. In the corrosion model, please explain a little more how the corrosion constants were deducted. For example, for nickel-plated, what is the Y-axis and units for figure 8-9

6. In line 22, it states "Table 7 and Figure 10 indicate that the values predicted by the model are generally larger than the actual values detected.". According to figure 10, the prediction values for N80 are smaller than actual values

Author's Response to Decision Letter for (RSOS-181899.R0)

See Appendix B.

RSOS-181899.R1 (Revision)

Review form: Reviewer 2

Is the manuscript scientifically sound in its present form?

Yes

Are the interpretations and conclusions justified by the results?

Yes

Is the language acceptable?

Yes

Is it clear how to access all supporting data?

Yes

Do you have any ethical concerns with this paper?

No

Have you any concerns about statistical analyses in this paper?

No

Recommendation?

Accept as is

Comments to the Author(s)

The revisions are acceptable! All my previous comments were addressed.

Decision letter (RSOS-181899.R1)

12-Mar-2019

Dear Professor Wang:

Title: Mechanism and Modelling of CO₂ Corrosion on Downhole Tools
Manuscript ID: RSOS-181899.R1

It is a pleasure to accept your manuscript in its current form for publication in Royal Society Open Science. The chemistry content of Royal Society Open Science is published in collaboration with the Royal Society of Chemistry.

RSC Associate Editor:
Comments to the Author:
(There are no comments.)

RSC Subject Editor:
Comments to the Author:
(There are no comments.)

Reviewer(s)' Comments to Author:
Reviewer: 2

Comments to the Author(s)
The revisions are acceptable! All my previous comments were addressed.

Appendix A

Manuscript Number: RSOS-181899 entitled: Mechanism and Modelling of CO₂ Corrosion on Downhole Tools, describes the corrosion failure in metal equipment and downhole tools used in oil and gas exploration owing to presence of corrosive environment of CO₂.

The study analyzed the composition, lattice, and crystalline forms of corrosion products in order to investigate the corrosion mechanisms. A CO₂ corrosion model was established.

It is a nice work and valuable for application in oil and gas industries. The paper is recommended for acceptance. The minor corrections are detailed below:

- i) Table 6,7: Give unit of corrosion rate.
- ii) Fig. 2: the 'y' axis legend mm/annum may be better as mmpy (mm/year).
- iii) Fig. 5: the 'x' axis legend is not clear.
- iv) Fig. 5: Can corrosion rate from Tafel plots be measured and compared with gravimetric weight-loss?
- v) The authors may include the following article in the introduction: Carbon dioxide Corrosion Inhibitors – A review, Arab J Sci Eng (2018) 43:1–22.

Appendix B

Dear Editor:

Thanks a lot for your letter concerning our new version of manuscript entitled “Mechanism and Modelling of CO₂ Corrosion on Downhole Tools” to (Manuscript ID: RSOS-181899).

Thanks a lot for your letter and for the reviewers’ comments concerning our previous version of manuscript entitled “Mechanism and Modelling of CO₂ Corrosion on Downhole Tools” to (Manuscript ID: RSOS-181899), too.

Those comments are all valuable and very helpful for revising and improving our paper, as well as the important guiding significance to our researches. We have studied comments carefully and have made correction which we hope meet with approval.

We appreciate for Editors/Reviewers’ warm work earnestly, and hope that the correction will meet with approval. Once again, thank you very much for your comments and suggestions.

Yours sincerely,

Jigang Wang; Lihui Meng

Northeast Petroleum University

The main corrections in the paper and the responds to the reviewer's comments are as flowing:

Responds to the reviewer's comments:

#Reviewer: 1

i) Response to comment: Table 6,7: Give unit of corrosion rate.

Response: We are very sorry for our inattention. We have made supplement according to the Reviewer's comments. In table 6,7: unit of corrosion rate is 'mm/year'.

ii) Response to comment: Fig. 2: the 'y' axis legend mm/annum may be better as mmpy (mm/year).

Response: Although 'mm/annum' has the same meaning with 'mm/year', we have made correction according to the Reviewer's comments. The 'y' axis legend mm/year is showed in Fig. 2.

iii) Response to comment: Fig. 5: the 'x' axis legend is not clear.

Response: We are very sorry for this. We have made 'x' axis legend clear according to the Reviewer's comments.

iv) Response to comment: Fig. 5: Can corrosion rate from Tafel plots be measured and compared with gravimetric weight-loss?

Response: Theoretically, according to the parameters of the Tafel plots, the corrosion rate can be seen, and the corrosion gravimetry can be obtained by multiplying the corrosion rate by the corrosion time. However, the corrosion of the four materials was mainly analyzed by the characterization of the Tafel plots. By comparing the degree of corrosion with the current density, it is finally verified that the nickel-plated material has the least corrosion degree.

#Reviewer: 2

1. Response to comment: In "experimental procedures", what is the film removing solution? What is it for?

Response: The main components of film removing solution: Hexamethylenetetramine 1g, Potassium-thiocyanate 1g, Nitric acid 52.5mL, Aniline 1mL, and Distilled water 444.5mL. The main function of the film removing solution is to clean. We have made supplement according to the Reviewer's comments.

2. Response to comment: Define corrosion rate. In line 21-23, 5 MPa was picked as the threshold pressure distinguishing between low and high corrosion rate. But

comparing the corrosion rate at 5 MPa and 8 Mpa, there is not much difference according to figure 2.

Response: First, it can be seen from the four graphs shown in figure2 that the corrosion rate keeps increasing from 5Mpa to 8Mpa, and the rate change of corrosion rate at 2a (20°C) and 2b (50°C) is also increasing. The results show that there is a qualitative change in the kinetics of CO₂ corrosion around 60 °C. The solubility of ferrous carbonate (FeCO₃) has a negative temperature coefficient, which decreases with increasing temperature. Therefore, between 60-110 °C, a corrosion-protective film layer can be formed on the steel surface, thus causing a transition in corrosion rate. Zone, so the rate change of 2c (80 ° C) and 2d (110 ° C) corrosion rate will not change significantly.

3. Response to comment: For Figure 4 energy spectra, provide some detailed on the spectra, especially the difference between a), b), and c). What are they referring to?

Response: Figure 4 energy spectra a)、b)、c) is the elemental analysis of three products with different corrosion degrees respectively, and the product analysis is carried out by analyzing the elements. We have made supplement according to the Reviewer's comments.

4. Response to comment: For table2-6, the definitions for severe vs minor vs strong corrosion are not clear?

Response: NACE rp0775-2005 (preparation and installation of corrosion hooks in oil field production and analysis of test data) specifies the degree of CO₂ corrosion in detail, and the specific content is shown below:

Table 1 regulations on the degree of CO₂ corrosion in the NACE

Classify	Uniform corrosion rate (mm/a)	pitting corrosion rate (mm/a)
Tiny corrosion	<0.025	<0.127
Minor corrosion	0.025-0.125	0.127-0.201
Severe corrosion	0.126-0.254	0.202-0.381
Strong corrosion	>0.254	>0.381

We have made supplement(Appendix I : regulations on the degree of CO₂ corrosion) according to the Reviewer's comments.

5. Response to comment: In the corrosion model, please explain a little more how the corrosion constants were deducted. For example, for nickel-plated, what is the Y-axis and units for figure 8-9?

Response: The corrosion constants were not deducted. It is only through fitting the

experimental data to choose the best constant. The value of a for N80 shows a linear trend, so the fitted value was calculated by $a_{N80}=(a_1+a_2)/2$. The value of a for the nickel-plated material fluctuates around the average, so this value, $a = 0.589$, was applied to the model. The Y-axis for figure 8-9 is a constant, and units are $\times 10^{-1}$. We are very sorry for our incorrect writing. We have made correction according to the Reviewer's comments. The units of figure 8-9 is supplemented in the paper.

6. Response to comment: In line 22, it states "Table 7 and Figure 10 indicate that the values predicted by the model are generally larger than the actual values detected.". According to figure 10, the prediction values for N80 are smaller than actual values?

Response: Thank you for correcting my inaccuracy. The prediction results suggest a corrosion rate slightly higher than the actual corrosion rate for nickel-plated material. For the highly corrosive N80 material, the predicted value is relatively small. Compared with the nickel-plated material, the single factor of the experimental condition is more prominent than the actual condition, so the predicted value is relatively small. We are very sorry for our incorrect writing. We have made correction according to the Reviewer's comments. This has been modified in the conclusion of the original article.